# Combined Prediction Model of Gas Concentration Based on Indicators Dynamic Optimization and Bi-LSTMs

**DOI:** 10.3390/s23062883

**Published:** 2023-03-07

**Authors:** Yujie Peng, Dazhao Song, Liming Qiu, Honglei Wang, Xueqiu He, Qiang Liu

**Affiliations:** 1School of Civil and Resource Engineering, University of Science and Technology Beijing, Beijing 100083, China; 2Key Laboratory of Ministry of Education for High-Efficient Mining and Safety of Metal, University of Science and Technology Beijing, Beijing 100083, China; 3State Key Laboratory of Coking Coal Exploitation and Comprehensive Utilization, Pingdingshan 467000, China; 4College of Safety Engineering, North China Institute of Science and Technology, Langfang 065201, China

**Keywords:** gas concentration, combined prediction model, indicators dynamic optimization, Bi-LSTMs, gas abnormal emission

## Abstract

In order to accurately predict the gas concentration, find out the gas abnormal emission in advance, and take effective measures to reduce the gas concentration in time, this paper analyzes multivariate monitoring data and proposes a new dynamic combined prediction method of gas concentration. Spearman’s rank correlation coefficient is applied for the dynamic optimization of prediction indicators. The time series and spatial topology features of the optimized indicators are extracted and input into the combined prediction model of gas concentration based on indicators dynamic optimization and Bi-LSTMs (Bi-directional Long Short-term Memory), which can predict the gas concentration for the next 30 min. The results show that the other gas concentration, temperature, and humidity indicators are strongly correlated with the gas concentration to be predicted, and Spearman’s rank correlation coefficient is up to 0.92 at most. The average *R*^2^ of predicted value and real value is 0.965, and the average prediction efficiency *R* for gas abnormal or normal emission is 79.9%. Compared with the other models, the proposed dynamic optimized indicators combined model is more accurate, and the missing alarm of gas abnormal emission is significantly alleviated, which greatly improves the early alarming accuracy. It can assist the safety monitoring personnel in decision making and has certain significance to improve the safety production efficiency of coal mines.

## 1. Introduction

A gas disaster is one of the most serious disasters in the coal mining process. Gas explosions, coal and gas outbursts, and other gas disasters will not only cause serious loss of life and property but also seriously affect mine safety production [1,2,3,4]. On 27 November 2014, a major gas explosion accident occurred in Songlin Coal Mine in Guizhou Province of China, resulting in 11 deaths and 8 injuries. This accident was caused by the accumulated gas in the roadway encountering frictional sparks from the fan blades running. The keys to preventing such accidents are to accurately predict the gas concentration and take effective measures to reduce the gas concentration in time [5,6].

The safety monitoring system, as one of the six major systems of the mine, provides real-time monitoring of gas concentration, wind speed, and other indicators in the mining working face [7]. The safety monitoring system is generally connected to the mine power supply system, which will automatically cut off the power supply once any gas abnormal emission is detected [8]. However, the start-up and shutdown of the power supply system will seriously affect the productivity of the coal mine. If the gas concentration can be predicted, the relevant measures to reduce the gas concentration can be taken in advance [9]. The multivariate monitoring data in the safety monitoring system provides conditions for the gas concentration prediction. The dynamic and accurate prediction of gas concentration can provide a decision-making basis for safety monitoring personnel, which is also significant to improve the safety production efficiency of coal mines.

In recent years, scholars have conducted in-depth research on the gas concentration prediction and put forward various prediction methods. Traditional prediction methods include gray prediction [10], chaotic time series [11], ARIMA time series prediction [12], etc. These methods make full use of the time series characteristics of gas concentration, but the prediction accuracy needs to be further improved. In latest years, with the gradual application of the neural network algorithm, many scholars have used it in gas concentration prediction, including long short-term memory network [13,14,15,16], recurrent neural network [17,18], and random neural networks [19], etc. As a deep learning technology, neural network has strong adaptability in time series prediction. It can rely on its special structural units to effectively use historical series information to realize deep mining of potential correlations between data, thus improving the accuracy of gas concentration prediction [13,20]. To further improve the prediction accuracy, scholars have made further improvements in terms of input data and model combinations. Cheng et al. [21] proposed an Evolutionary attention-based temporal graph convolutional network (EAT-GCN) to simultaneously capture the spatial and temporal dependences. Wang et al. [22] proposed the LSTM-LightGBM model with a variable weight combination method based on residual assignment. Wu et al. [23] proposed a model that integrates the t-distributed stochastic neighbor embedding algorithm (t-SNE) and the support vector regression (SVR) algorithm. Meng et al. [24] proposed a novel prediction method that combines classical time series analysis with these deep learning models. Dey et al. [25] proposed the t-SNE_VAE_bi-LSTM prediction model that combines the t-SNE, VAE, and bi-LSTM networks. Xu et al. [26] constructed a new IWOA-LSTM-CEEMDAN prediction model based on the improved whale optimization algorithm (IWOA). The above methods have improved the prediction accuracy.

In order to further improve the prediction accuracy of gas concentration, this paper analyzes the multivariate monitoring data and proposes a new combined prediction model of gas concentration based on indicators dynamic optimization and Bi-LSTMs. The proposal of this new prediction model significantly improves the accuracy of gas concentration prediction and also has a certain significance to improve the safety production efficiency of coal mines.

## 2. Research and Theoretical Background

### 2.1. Data Sources

The research data in this paper are obtained from IJCRS’15 Data Challenge: Mining Data from Coal Mines, organized at the Knowledge Pit web platform [27,28,29,30]. These data are the multivariate monitoring data near the long wall of the fully mechanized mining face in Poland, mainly including gas concentration, wind speed, temperature, humidity, air pressure, working current of the fully mechanized mining machine, etc. The layout of various sensors is shown in Figure 1. In Figure 1, sensors descriptions are as follows: AN-anemometer [m/s] (including AN311, AN422, and AN423); TP-temperature [°C] (including TP1711 and TP1721, the sensor type of TP1711 is temperature THP, three-component sensor THP2/94, and the sensor type of TP1721 is temperature THP, three-component sensor THP2/93); RH-humidity [%RH] (including RH1712 and RH1722, the sensor type of RH1712 is temperature THP, three-component sensor THP2/94, and the sensor type of RH1722 is temperature THP, three-component sensor THP2/93); BA-barometer [hPa] (including BA1713 and BA1723); MM-methane meter [%CH_4_] (including MM252, MM261, MM262, MM263, MM264, MM256 and MM211); CM861-high concentration methane meter [%CH_4_], the range is 0–100; TC862-temperature inside the pipeline [°C], the range is 10–40; WM868-methane delivery calculated according to CM861, CR863, P, TC862 [m^3^/min], the range is 0–50; CR863-sensor for pressure difference on the methane drainage flange [Pa], the range is 0–250; P-pressure inside the methane drainage pipeline [kPa], the range is 0–110; AMP1-current of the left cutting head of the cutter loader [A]; AMP2-current of the right cutting head of the cutter loader [A]; DMP3-current of the left haulage in the cutter loader [A]; DMP4-current of the right haulage in the cutter loader [A]; AMP5-current of the hydraulic pump engine in the cutter loader [A]; F-driving direction, 1 = left, 0 = right; V-cutter loader speed [Hz], V_min_ = 3 Hz, V_max_ = 100 Hz. Herz values are then transformed into m/min, 100 Hz equal to about 20 m/min.

The fully mechanized mining machine moves along the long wall between MM262 and MM264. The larger the working current of the coal cutter, the higher efficient the coal cutting will be, which theoretically leads to more gas diffusion into the mined-out area, and the monitoring value of the gas sensor will increase accordingly. The three key sensors, MM263, MM264, and MM256, are located near the upper corner of the fully mechanized mining face, which is prone to gas accumulation. The three sensors can monitor the change of gas concentration at this location in real-time, and the coal cutter will automatically shut down when the gas concentration of any sensors reaches the alarm threshold (the alarm threshold value of the three sensors is 1.0%). If the gas abnormal emissions can be predicted in advance, it is possible to reduce the coal cutter’s cutting speed, i.e., to give more time for the gas to diffuse before the entire line is stopped. Therefore, it is particularly important to predict the gas concentration of MM263, MM264, and MM256.

### 2.2. Prediction Theories

#### 2.2.1. Spearman’s Rank Correlation Coefficient

In order to accurately predict the gas concentration near the fully mechanized mining machine, the Spearman’s rank correlation coefficient (SRC) is used to dynamically optimize prediction indicators. Spearman’s rank correlation coefficient (SRC) refers to the degree of correlation between indicators. The larger the SRC, the stronger the correlation between the two indicators, indicating that the indicator is useful for predicting the other indicator [31]. SRC is not affected by the overall distribution and sample size of data and is more suitable for non-normal distributed coal mine time series data [32]. Suppose that the two variables are *P* and *Q* respectively, and the number of their elements is both *N*. *P_i_* and *Q_i_* represent the *i*th (1 ≤ *i* ≤ *N*) value of *P* and *Q*, respectively, and the two variables are sorted (both in ascending or descending order) to obtain the two variable ranking sets *p* and *q*, where the elements *p_i_* and *q_i_* represent the ranking of *P_i_* and *Q_i_* in *P* and *Q*, respectively. A ranked difference set *d* is obtained by subtracting the corresponding elements in the sets *p* and *q*, where *d_i_* = *p_i_* − *q_i_* (1 ≤ *i* ≤ *N*), and the SRC between the two variables *P* and *Q* is the following:(1)$$\rho =1-\frac{6\sum_{i=1}^{N}d_{i}^{2}}{N\left(N^{2}-1\right)}$$

If the SRC is 0.8~1.0, there is an extremely strong correlation between variables; 0.6~0.8, strong correlation; 0.4~0.6, moderate correlation; 0.2~0.4, weak correlation; 0.0~0.2, very weak correlation or no correlation. When the SRC is a negative, there is negative correlation between variables.

#### 2.2.2. Long Short-Term Memory Network

Bi-directional Long Short-term Memory (Bi-LSTM) and Long Short-Term Memory (LSTM) networks are applied to establish a combined prediction model of gas concentration. Long Short-Term Memory (LSTM) network is an improved recurrent neural network (RNN) [33,34]. Three logic control units, namely, input gate, output gate, and forget gate, are added to the basic structure of RNN to control the input and output of information flow and the state of the cell units by setting the weights at the edges where the neural network memory cells are connected to other parts. LSTM overcomes the shortcomings of RNN, which cannot deal with long-distance dependence and is prone to gradient disappearance and explosion. The basic structure of LSTM is shown in Figure 2.

*C_t_* and *h_t_* represent the memory state and the hidden state of LSTM, respectively, and *x_t_* represents the input of the model, *σ* is a sigmoid activation function. *F_t_*, *i_t_*, and ${\tilde{C}}_{t}$ calculation formula are the following:(2)$$f_{t}=\sigma \left(W_{f}\left[h_{t-1},x_{t}\right]+b_{f}\right)$$
(3)$$i_{t}=\sigma \left(W_{i}\left[h_{t-1},x_{t}\right]+b_{i}\right)$$
(4)$${\tilde{C}}_{t}=\tanh\left(W_{c}\left[h_{t-1},x_{t}\right]+b_{c}\right)$$
(5)$$O_{t}=\sigma \left(W_{o}\left[h_{t-1},x_{t}\right]+b_{o}\right)$$
(6)$$h_{t}=O_{t}\times \tanh\left(C_{t}\right)$$
where tanh is the hyperbolic tangent activation function; *W_f_*, *W_i_*, *W_c,_* and *W_o_* are the weight matrices connected to the input vector *x_t_* and the previous hidden state *h_t_*_−1_ for each layer, respectively; *b_f_*, *b_i_*, *b_c_* and *b_o_* are the bias terms of each layer, respectively.

#### 2.2.3. Bi-Directional Long Short-Term Memory Network

Bi-directional Long Short-Term Memory (Bi-LSTM) network [35,36] combines the information of the input sequence in both forward and backward directions based on the LSTM, i.e., the sequence is input into LSTM in forward and reverse order, respectively. Bi-LSTM makes the feature acquired at moment *t* to have information between the past and the future, which can effectively ensure the accuracy of time sequence prediction [37,38]. For the output at moment *t*, the forward LSTM layer has the information at moment *t* in the input sequence and the previous moments, while the backward LSTM layer has the information at moment *t* in the input sequence and the moments after. The structure of Bi-LSTM is shown in Figure 3.

It can be seen from Figure 3 that Bi-LSTM is composed of two LSTMs. When solving, the hidden vector *h_t_*_−1_ of the front term is calculated to generate a new hidden vector ${\vec{h}}_{t}$, and the hidden vector *h_t_*_+1_ of the rear term is also calculated to generate a new hidden vector ${\overset{\leftarrow }{h}}_{t}$. Combine the output results of the positive and negative input sequences to obtain the result *Y_t_*, and the calculation formula is as follows:(7)$${\vec{h}}_{t}=\mathrm{LSTM}\left(x_{t},{\vec{h}}_{t-1}\right)$$
(8)$${\overset{\leftarrow }{h}}_{t}=\mathrm{LSTM}\left(x_{t},{\overset{\leftarrow }{h}}_{t-1}\right)$$
(9)$$Y_{t}=\tanh\left(W_{Y}\left[{\vec{h}}_{t}+{\overset{\leftarrow }{h}}_{t}\right]+b_{Y}\right)$$
where, *W_Y_* is the weight matrix of each layer connected to the previous hidden state *h_t_*_−1_; *b_Y_* is the bias term.

## 3. Methodology

### 3.1. Prediction Methods

Through the analysis of multivariate monitoring data, this paper proposes a new combined prediction method of gas concentration based on indicators dynamic optimization and Bi-LSTMs, as shown in Figure 4. First, the multivariate monitoring data is processed into time series data, and the SRC between the gas concentration to be predicted and other indicators within the time window length *l* are analyzed. Considering the SRC changes with time, the prediction indicators are dynamically optimized. Then, the time series and spatial topology features of the optimized indicators are extracted to build the feature matrix *X*, which is input into the Bi-LSTM single indicator model to obtain the prediction result *Y*. After that, each optimized single indicator prediction result is input into the LSTM combined model again and obtain the gas concentration prediction value *Y_predict_*. Finally, the model is continuously evaluated and optimized to improve the applicability of the prediction model.

### 3.2. Prediction Model

#### 3.2.1. Correlation Analysis

The safety monitoring system stores all kinds of monitoring data, but not all indicators are useful for gas concentration prediction, and some may even be counterproductive. Redundant indicators will prevent the model from finding potential laws of data, cause dimensional disasters, and reduce the prediction efficiency of the model [20]. SRC shows the degree of correlation between indicators. Take MM256 as an example, set the length of time window to 1 day, calculate the SRC of MM256 and other indicators, and select the top three indicators of the strongest correlation with MM256 for 7 consecutive days, as shown in Table 1.

As can be seen from Table 1, the top three indicators of the strongest correlation with MM256 are almost different every day. Most of the three indicators contain a gas indicator, that is, the other gas indicators are mostly useful for MM256 prediction. However, the indicator with strongest correlation is not necessarily the gas indicator. For example, on the 26th and 27th days, the indicator with strongest correlation is RH1712, and the SRC is more than 0.7, showing a strong correlation. The SRC between each indicator and MM256 also changes every day. In order to clarify the SRC between other indicators and MM256, the frequency of each indicator in the top three indicators is counted, as shown in Figure 5.

It can be seen from Figure 5 that the frequency of each indicator is different, but the frequency of gas indicators is higher than other indicators on average, which also shows that gas indicators are mostly useful for MM256 prediction. In addition, humidity indicators (RH1212, RH1722) and temperature indicators (TP1721, TP1711, TC862) also appear more frequently, indicating that MM256 is more influenced by temperature and humidity. Among the top three indicators of the strongest correlation, MM263, RH1712, and MM264 have the highest frequency. Therefore, take these three indicators as an example to calculate the SRC with time, as shown in Figure 6.

As can be seen from Figure 6, the SRC of each indicator fluctuates drastically with time, and the highest SRC of three indicators all reach above 0.9, which is an extremely strong correlation. The extreme differences are 1.53, 1.1, and 1.22, respectively, which indicates that each indicator’s prediction effect on MM256 is unstable and the effect changes with time. Therefore, it is difficult to apply fixed one or some indicators to achieve stable and accurate prediction.

#### 3.2.2. Indicators Dynamic Optimization

From the correlation analysis in Section 3.2.1, it can be seen that gas concentration, humidity, and temperature indicators are highly correlated with MM256. However, the top three indicators are not fixed, and the SRC of each indicator fluctuates with time. Analyze the above reasons, the underground environment of the coal mine is complex, and the gas concentration is disturbed by various factors (such as geological, human, equipment factors, etc.). In a certain period of time, the influence of various factors on gas concentration is different. For a certain factor, its influence on the gas concentration is not fixed, so it is difficult to determine its relationship with the gas concentration. Similar results are obtained when analyzing MM263 and MM264 data. It is more important to consider the correlation between gas concentration and other indicators at the prediction moment. Therefore, a dynamic optimization method for prediction indicators based on SRC is proposed, as shown in Figure 7.

For the multivariate time series monitoring data, calculate the SRC of each indicator with the gas indicator to be predicted within the time window length *l* of moment *t_c_*, and obtain *ρ* = (*ρ*_1_, *ρ*_2_, …, *ρ_n_*). According to the SRC, the indicators with moderate correlation and above are optimized, i.e., the SRC *ρ*′ = (*ρ*′_1_, *ρ*′_2_, …, *ρ*′*_m_*) is higher than 0.4, those indicators form the optimized indicators set. If there are less than 2 indicators in the optimized indicators set, i.e., except for the gas indicators to be predicted, no other indicators’ SRC is higher than 0.4, then the five indicators corresponding to the 5 largest SRC in *ρ* = (*ρ*_1_, *ρ*_2_, …, *ρ_n_*), gas concentration, temperature and humidity indicators form the optimized indicators set. Among them, gas concentration indicators include MM252, MM261, MM262, MM263, MM264, MM256, MM211 and CM861, recorded as MM; temperature indicators include TP1721, TP1711 and TC862, recorded as T; humidity indicators include RH1212 and RH1722, recorded as H. This method not only considers the spatiotemporal relationship of indicators but also the change of correlation between other indicators and gas concentration with time.

#### 3.2.3. Features Extraction

Feature extraction is a very critical step in prediction model design, and the quality of features directly affects the model prediction performance. Time series and spatial topology features are extracted from the optimized indicators.

(1)Time series features

The monitoring data are typical time series data, and the gas time series is set as follows:(10)$$C\left(t\right)=\left\{C_{1}\left(t_{1}\right),C_{2}\left(t_{2}\right),C_{3}\left(t_{3}\right),\cdot \cdot \cdot ,C_{n}\left(t_{n}\right)\right\}$$
where, *t_i_* is time; *C_i_*(*t_i_*) is the gas concentration at moment *t_i_*; *n* is the time length of gas time series.

Calculate the features of the gas time series *C*(*t*):

Gas concentration monitoring value *C_c_*(*t_c_*) at the moment *t_c_*; First order difference value of gas concentration at the moment *t_c_*, *D_c_*(*t_c_*) = *C_c_*(*t_c_*) − *C_c_*_−1_(*t_c_*_−1_); Statistical features of the gas concentration time series *C_l_*(*t*) = {*C_c_*_−*l*_(*t_c_*_−*l*_), …, *C_c_*_−2_(*t_c_*_−2_), *C_c_*_−1_(*t_c_*_−1_), *C_c_*(*t_c_*)} with time window length *l* before moment *t_c_*, including 10 statistical features of maximum, average, root mean square, variance, standard deviation, dispersion coefficient, peak factor, skewness, kurtosis and range of *C_l_*(*t*) [39].

(2)Spatial topology features

During the production process of a coal mine, gas flows into the roadway from fallen coal, coal wall, and mined-out area, and the gas continuously diffuses in the roadway. According to the correlation analysis in Section 3.2.1, there is a strong correlation between different indicators at different monitoring points. Randomly select the time series monitoring data of different sensors with a time length of 1 day, as shown in Figure 8.

As shown in Figure 8, The monitoring values of different sensors are correlated in time and space, and their time series data show a similar trend, but the peak value shows a certain time lag, which is obviously related to sensor layout. Therefore, when predicting the gas concentration at a certain measuring point, in addition to calculating its own time series features, it also calculates the time series features of the optimized indicators and inputs them into the model as spatial topology features.

#### 3.2.4. Gas Concentration Prediction Model

(1)Bi-LSTM single indicator model

Indicators dynamic optimization is applied to predict gas concentration, and single indicator prediction model is established based on single indicator time series data. Set the single indicator time series as follows:(11)$$X\left(t\right)=\left\{x_{1}\left(t_{1}\right),x_{2}\left(t_{2}\right),x_{3}\left(t_{3}\right),\cdot \cdot \cdot ,x_{n}\left(t_{n}\right)\right\}$$
where *t_i_* is the time; *x_i_*(*t_i_*) is the monitoring value at the moment *t_i_*; *n* is the time length of the single indicator time series. According to the features extraction method in this paper, calculate the time series features of the sequence *X_l_*(*t*) = {*X_c_*_−*l*_(*t_c_*_−*l*_), …, *X_c_*_−2_(*t_c_*_−2_), *X_c_*_−1_(*t_c_*_−1_), *X_c_*(*t_c_*)} with the following time window length *l* before moment *t_c_*:(12)$$X_{t}{}_{{}_{c}}=\left[x_{1}^{t_{c}},x_{{}_{2}}^{t_{c}},x_{3}^{t_{c}},\cdot \cdot \cdot ,x_{{}_{12}}^{t_{c}}\right]$$

Then the features matrix with lag step of *h* and prediction step of *p* is the following:(13)$$X={\left[X_{t_{c}-hp+p},\cdot \cdot \cdot ,X_{t_{c}-p},X_{t_{c}}\right]}^{T}$$
i.e.,
(14)$$X=\left[\begin{array}{c} \begin{array}{cccc} x_{1}^{t_{c}-hp+p} & x_{2}^{t_{c}-hp+p} & \ldots & x_{12}^{t_{c}-hp+p} \end{array}\\ \ldots \\ \begin{array}{cccc} x_{1}^{t_{c}-p} & x_{2}^{t_{c}-p} & \ldots & x_{12}^{t_{c}-p} \end{array}\\ \begin{array}{cccc} x_{1}^{t_{c}} & x_{2}^{t_{c}} & \ldots & x_{12}^{t_{c}} \end{array} \end{array}\right]$$

Different features have different dimensions and units, which will affect the model prediction accuracy. Therefore, firstly, the features matrix *X* is mapped between [0, 1] by using the method of MinMaxScaler_fit_transform to obtain the standardized feature matrix *X_N_*. Input *X_N_* into Bi-LSTM, and then input its output results into 3 layers DenseNet. Feature reuse is achieved through the connection of features on the channel, and the activation function RELU is used to de-linearize between DenseNet layers. The Dropout layer is set to discard neurons from the network with a probability of 20% to prevent overfitting of the model. Dense connections with weight sharing are used to filter out process noise and interference information. Different features are learned in a supervised manner to output predicted values. Finally, the maximum gas concentration *Y* within *T* time after the moment *t_c_* is obtained through MinMaxScaler_inverse_transform. *T* is defined as the prediction length. Bi-LSTM single indicator model network structure is shown in Figure 9.

(2)LSTM dynamic optimized indicators combined model

Gas disasters are often the result of multiple factors coupling, and combined prediction can excavate more hidden information. The simple combination of indicators does not necessarily improve the prediction accuracy, so it is necessary to optimize and process the information from various indicators. Bi-LSTM single indicator model can predict the maximum value *Y* of gas concentration in time *T* after the moment *t_c_*. If *Y* is larger than the threshold, the gas emission is abnormal in time *T*, and corresponding measures should be taken at once. By applying the proposed dynamic optimization method, *m* prediction indicators at moment *t_c_* are optimized. According to Formulas (11)–(14), the features matrix X = [*X*_1_, *X*_2_, …, *X_m_*] with the lag step of *h* and the prediction step of *p* are calculated respectively. The features matrix of each indicator is input as the Bi-LSTM single indicator model, and obtain the prediction result Y = [*Y*_1_, *Y*_2_, …, *Y_m_*].

Combined prediction is to combine the results of different prediction models to improve the prediction accuracy as much as possible [40]. In this paper, the prediction result Y = [*Y*_1_, *Y*_2_, …, *Y_m_*] is combined and standardized to obtain Y*_N_*, input Y*_N_* into the LSTM of the two layers in turn, and the Dropout layer is also set to discard neurons from the network with a 20% probability. DenseNet layer connection and weight sharing are adopted. The final prediction value *Y_predict_* of gas concentration in *T* time after the moment *t_c_* is obtained through MinMaxScaler_inverse_transform. LSTM-optimized indicators combined model network structure are shown in Figure 10.

### 3.3. Model Evaluation

In order to quantitatively evaluate the performance of the prediction model, evaluation indexes are used to evaluate the prediction results. The errors are divided into longitudinal error and transverse error. Longitudinal error is used to analyze the long-term operation of system in amplitude, while transverse error is utilized to study the prediction performance in time delay [40]. Longitudinal error includes mean absolute error (*MAE*), root mean square error (*RMSE*), etc.; transverse error includes determination coefficient (*R*^2^), etc. The closer the longitudinal error is to 0, and the closer the transverse error is to 1, the better the prediction model performance is. These evaluation indexes are defined as follows:(15)$$MAE=\frac{1}{n}\sum_{i=0}^{n}\left|y_{i}-{\hat{y}}_{i}\right|$$
(16)$$RMSE=\sqrt{\frac{1}{n}\sum_{i=0}^{n}{\left(y_{i}-{\hat{y}}_{i}\right)}^{2}}$$
(17)$$\begin{array}{cc} R^{2}=1-\frac{\sum_{i=1}^{n}{\left(y_{i}-{\hat{y}}_{i}\right)}^{2}}{\sum_{i=1}^{n}{\left(y_{i}-\bar{y}\right)}^{2}} & \bar{y}=\frac{1}{n} \end{array}\sum_{i=1}^{n}y_{i}$$
where, *n* is the total number of samples; *y_i_* is the real value of gas concentration; ${\hat{y}}_{i}$ is the predicted value of gas concentration.

According to the predicted value of gas concentration and the threshold value (1.0%), it can be inferred whether the gas emission is abnormal. The gas emission situations are divided into the following two categories: gas abnormal emission and gas normal emission. Considering the influence of sample imbalance [41], missing alarm and false alarm comprehensively, the false alarm rate (*FAR*), missing alarm rate (*MAR*), and prediction efficiency (*R*) [39,42,43] are applied to objectively evaluate the prediction model for gas abnormal or normal emission. These evaluation indexes are defined as follows:(18)$$FAR=\frac{FP}{FP+TN}$$
(19)$$MAR=\frac{FN}{FN+TP}$$
(20)$$R=\frac{TP}{TP+FN}-\frac{FP}{FP+TN}$$
where *TP* is the number of samples with both predicted and real values exceeding the threshold; *FN* is the number of samples with real values exceeding the threshold but not the predicted value, i.e., the number of missing alarm samples; *FP* is the number of samples with predicted value exceeds the threshold but not the real value, i.e., the number of false alarm samples; *TN* is the number of samples with neither predicted value, nor real value exceeds the threshold.

## 4. Results

### 4.1. Data Preprocessing

The data provided by the Knowledge Pit web platform includes a training set and a test set. The training set contains the monitoring values of 28 sensors, which are given in time order, with each type of sensor producing one monitoring value per second, but there is a time overlap in the training set. The test set has the same data format as the training set, and its time does not overlap, but it is generated in random order. Therefore, this paper only analyzes the training set data. The time window length of the training set is 10 min, and the sliding step length is 1 min. The training set overlapping data is processed into time series data, as shown in Figure 11.

### 4.2. Model Training

The time overlapping test set data is processed through the method as shown in Figure 11, and 36 days of multivariate monitoring data of fully mechanized mining face is obtained. Set the time window length *l* = 1 d, lag step length *h* = 5, prediction length *T* = 30 min, that is, according to the multivariate monitoring data with 1 day before the current moment, calculate the features matrix of optimized indicators with the length of 5, and predict gas abnormal or normal emission within 30 min after the current moment. The prediction step *p* = 30 min, that is, it is predicted once every 30 min.

For each optimized indicator, according to the feature extraction and sample construction methods in this paper, 1629 sample datasets between features matrix *X* and *Y* corresponding one by one are constructed with the help of the Python-Keras framework. The first 80% of the sample datasets are divided into the training set and the last 20% into the test set. Input the training set and test set into the Bi-LSTM single indicator model, respectively, and the model is continuously trained and optimized. In the model training process, the Nadam optimizer is used to train the network, Epochs = 50, batch_size = 32, and *lr* = 0.001. For the prediction result *Y* of the Bi-LSTM single indicator model, the sample datasets corresponding to *Y* and *Y_predict_* are constructed, which are input into the LSTM optimized indicators combined model again. In order to avoid overfitting, 5-fold cross-validation is used to adjust the hyperparameters for automatic optimization. The model training is reflected by cross-entropy loss (Loss), which is defined as follows:(21)$$Loss=-\frac{1}{n}\sum_{i=1}^{n}\left[y_{i}\log{\hat{y}}_{i}+\left(1-y_{i}\right)\log\left(1-{\hat{y}}_{i}\right)\right]$$

For the three key sensors MM256, MM263, and MM264, the gas indicator to be predicted, all-indicators, specified indicators (gas concentration, temperature, and humidity indicators, i.e., MM, T, and H), dynamically optimized indicators are respectively established in the following four models: Bi-LSTM single indicator model, Bi-LSTMs all-indicators combined model, Bi-LSTMs specified indicators combined model, and Bi-LSTMs dynamic optimized indicators combined model. The changes of Loss with Epoch during the training of different models are shown in Figure 12.

As can be seen from Figure 12, with the increase in Epoch, the Loss fluctuates and decreases during the model training. When Epoch = 10, the Loss gradually tends to be flat, and the model convergence performance is better. The convergence of the single indicator model is slightly poor, with large fluctuations. The all-indicators model has inferior convergence performance than the specified indicators model and the optimized indicators model due to the introduction of more redundant indicators. The specified indicators model is better than the optimized indicators model and converges slightly faster for MM256, but when Epoch = 30, the convergence of the two models is consistent. The optimized indicators model converges significantly better than the specified indicators model for MM263 and MM264. Indicators dynamic optimization is helpful to improve the convergence performance of the model, which verifies the necessity of indicators dynamic optimization to determine the input indicators of the model. Meanwhile, the changes of Loss with Epoch show that the Bi-LSTMs dynamic optimized combined indicators model is better for gas concentration prediction, and the prediction results of which can reflect the gas abnormal or normal emission in the future.

### 4.3. Prediction Results

Using the trained Bi-LSTMs dynamic optimized combined indicators model, 325 groups of test set data are used for model testing, and the prediction results are shown in Figure 13. It can be seen from Figure 13 that the fluctuation trend of the predicted value is basically consistent with the real value, and the proposed model has good prediction performance for gas concentration. In the prediction of gas abnormal emission, there is no false alarm for the three key sensors, but there are a few missing alarms.

### 4.4. Model Comparison

In order to comprehensively evaluate the prediction performance of the model, the predicted and real value of gas concentration, gas emission prediction, and actual gas emission situation of different models are compared. According to Equations (15)–(17), the *R*^2^, *MAE*, and *RMSE* are applied to evaluate the model’s gas concentration prediction performance. According to the gas concentration prediction value, the threshold value (1.0%), and Equations (18)–(20), the *FAR*, *MAR*, and *R* are applied to evaluate the model’s gas emission prediction performance. The prediction results of different indicators models are shown in Figure 14 the gray area in Figure 14 indicates that the prediction of gas abnormal or normal emission events is accurate. The evaluation of prediction results is specifically shown in Table 2 and Table 3, and *R*^2^′, *MAE*′, *RMSE*′, *FAR*′, *MAR*′, and *R*′ represent the average values of the corresponding evaluation indexes.

Compare and analyze the prediction results in Figure 14, Table 2 and Table 3. As shown in Figure 14a, Bi-LSTM single indicator model has poor prediction performance for gas concentration, the prediction performance of MM256 and MM264 is slightly better than that of MM263. The average *R*^2^ of predicted value and real value is 0.667, and the average *MAE* and *RMSE* are 0.156 and 0.143, respectively. There is a large difference between the predicted value and the real value. The prediction performance of a single indicator on gas abnormal or normal emission is also poor, with a high missing alarm rate, among as high as 100.0% for MM263, and the average missing alarm rate is 67.2%, which is quite serious. At the same time, there are also a few false alarms, with an average false alarm rate of 1.2% and an average prediction efficiency *R* of 31.6%.

As shown in Figure 14b, the prediction performance of the Bi-LSTMs all indicators combined model for gas concentration is slightly better than that of the single indicator model, but the results are also unsatisfactory. The average *R*^2^ is 0.675, the average *MAE* and *RMSE* are 0.095 and 0.138, respectively, and the error between the predicted and real value is reduced compared with the single indicator model. However, the prediction performance of all indicators on gas abnormal or normal emission is inferior to that of a single indicator, and the missing alarm rate of MM263 is still 100.0%, and the average missing alarm rate is 70.8%, which is still serious instead of decreasing compared to the single indicator model. The false alarm rate is reduced, with an average of 0.4%, and the average prediction efficiency *R* of 28.7%, which is lower than that of the single indicator.

As shown in Figure 14c, the prediction performance of the Bi-LSTMs specified indicators combined model for gas concentration is similar to that of single indicator and all-indicators models. The average *R*^2^ is 0.687 and the average *MAE* and *RMSE* are 0.100 and 0.138, respectively. However, the prediction performance of gas abnormal or normal emission is better than that of single indicator and all-indicators models. The missing alarm rate of MM263 is reduced to 68.2%, and the average missing alarm rate is 58.8%, the missing alarm is still serious. Compared with all indicators, the average false alarm rate is still 0.4%. The average prediction efficiency *R* is 40.7%, which is higher than that of single indicator and all indicators models.

As shown in Figure 14d, the Bi-LSTMs dynamic optimized indicators combined model has a better prediction performance for gas concentration, *R*^2^ of the three sensors is all above 0.90, with an average value of 0.965, average *MAE* and *RMSE* is 0.039 and 0.046 respectively. Compared with the single indicator, all indicators, and specified indicators models, the average *R*^2^ is significantly improved. The prediction performance for gas abnormal or normal emission is also better than that of other models. It significantly reduces the missing alarm and eliminates false alarms, with a false alarm rate of 0.0% and an average missing alarm rate of 20.1%. The average prediction efficiency *R* is 79.9%, which is 152.8%, 178.4%, and 96.3% higher than that of applying the single indicator, all-indicators, or specified indicators models, respectively.

## 5. Discussion

Mining working faces are important production places of a coal mine, and safe and efficient coal mining is the key to coal mine production and operation. Gas concentration usually increases abnormally before the occurrence of gas disasters such as gas explosions and coal and gas outbursts. In order to prevent gas accidents and ensure safe production during coal mining, the Coal Mine Safety Regulations of China stipulate that the gas concentration in the coal mining working face shall not exceed 1%; otherwise, an alarm will be given. The setting of this threshold value has effectively reduced gas accidents. However, with the gradual increase in the mine depth, the ground stress, and gas content increase. When the gas concentration exceeds the threshold value, coal mining processes such as coal cutters will be interrupted, and the operators will also fall into panic, which seriously affects the safety production efficiency of the coal mine and the psychological health of the operators. Therefore, the dynamic and real-time prediction of the gas concentration can improve the working efficiency of the coal cutters and effectively ensure the safe and continuous production of the mining working face.

In the process of coal mining, gas flows into the roadway from the fallen coal, coal wall, and mined-out area, and the gas continuously diffuses in the roadway. In this process, the gas emission from the coal wall and adjacent coal seams is relatively stable, while the gas from the fallen coal and mined-out area is easy to accumulate, which is greatly influenced by the degree of coal fragmentation, coal cutter speed, wind speed, etc. The influences of this information are largely reflected in other types of sensors, such as gas, temperature, humidity, and air pressure sensors. However, not every sensor information is useful for gas concentration prediction, and the correlation between other sensors and gas concentration changes with time. Therefore, this paper proposes a dynamic optimization method for prediction indicators. This method not only considers the spatiotemporal relationship of indicators but also the change of correlation between other indicators and gas concentration with time. On this basis, establishes a combined prediction model of gas concentration based on indicators dynamic optimization and Bi-LSTMs, which further improves the accuracy of gas concentration prediction and its abnormal emission alarming. The prediction model is based on multivariate real-time monitoring data, which can realize real-time and dynamic prediction of gas concentration. However, the prediction model is more suitable for the prediction of gas or other indicators monitored by multiple sensors compared with a few sensors, which is also the limitation of the model. Admittedly, the alarming effect of the proposed prediction model is significantly better than that of the single indicator model, all-indicators combined model, and specified indicators combined model.

Analyzing the above reasons, the single indicator model only uses its own historical data for prediction, with less input information, which can only reflect the gas time features, but not the spatial topology features. Compared with the single indicator model, all-indicators combined model introduces more information for the model, but this information is not all useful for prediction, and some information may even have a negative effect, causing dimensional disasters. The specified indicators combined model considers the spatial topology features, and the prediction indicators are relatively better, but the quality of these indicators changes with time and is not optimal at the prediction moment. Therefore, compared with single indicator and all-indicators, the performance of the model is only slightly improved. The dynamic optimized indicators combined model optimizes strongly correlated indicators, and different moments correspond to different indicators, which ensures the optimal sample indicators at the prediction moment. Compared with a single indicator, all-indicators, and specified indicators, the model introduces the most useful information for prediction at the prediction moment. Therefore, indicators dynamic optimization significantly improves the performance of the model.

The prediction model established in this paper sets many adjustable parameters, including time window length *l*, lag step *h*, prediction length *T*, prediction step *p*, etc. In this paper, set *l* = 1 d, lag step *h* = 5, *T* = 30 min, and *p* = 30 min, calculate the features matrix of optimized indicators with the length of 5 based on the multivariate monitoring data with 1 d before the current moment, apply optimized indicators to predict the gas abnormal or normal emission within 30 min after the current moment and the prediction is made once every 30 min. Of course, these parameters can be adjusted according to the actual demand of the coal mine, but this may affect the prediction performance of the model. The influence of these parameters on model prediction performance will be also the focus of future research. In addition, this study is a “Point prediction” of gas concentration for three key gas sensors. We plan to expand the gas concentration prediction to “Two-dimension prediction”, or even “Three-dimension prediction”, so as to visualize the gas concentration prediction area in the mining face, which is also the key direction of future research. The research in this paper lays the foundation for future research.

## 6. Conclusions

Accurate prediction of gas concentration in the mining face and early detection of gas abnormal emissions are the keys to preventing gas accidents. In order to avoid the contingency of a single indicator early alarming method and capture the precursor information of gas disaster more accurately, this paper proposes a new combined prediction model of gas concentration based on indicators dynamic optimization and Bi-LSTMs, which is proved to be effective. The main conclusions are as follows:(1)The Spearman’s rank correlation between gas concentration and other indicators is analyzed. The gas concentration, temperature, and humidity indicators have a strong correlation with the gas concentration to be predicted, and Spearman’s rank correlation coefficient can reach 0.92 at most. For a certain indicator, its correlation with gas concentration fluctuates with time, with a maximum range of 1.53. The prediction effect of a single indicator on gas concentration is not stable, it is necessary to consider that the correlation changes with time at the prediction moment to dynamically optimize indicators;(2)A dynamic optimization method for prediction indicators based on Spearman’s rank correlation coefficient is proposed. This method not only considers the spatiotemporal relationship of indicators, but also the change of correlation between other indicators and gas concentration with time, reducing the training cost and model complexity, and improving the model convergence speed and training efficiency;(3)A new combined prediction model of gas concentration based on indicators dynamic optimization and Bi-LSTMs is established, which can predict the gas concentration for the next 30 min. Using the proposed model, the average *R*^2^ of predicted value and real value is 0.965, and the average prediction efficiency *R* for gas abnormal or normal emission is 79.9%. The proposal of this new prediction model significantly reduces the missing alarm and eliminates false alarms. Compared with other models, the gas concentration prediction is more accurate, and the missing alarm of gas abnormal emission is greatly alleviated, which greatly improves the alarming accuracy;(4)The research of this paper is a “Point prediction” of gas concentration for three key gas sensors. “Two-dimension prediction” or “Three-dimension prediction” for gas concentration could realize to visualize the gas concentration prediction area in the mining face, and it is also the key direction of future research. In addition, the influence of adjustable parameters of the model on prediction performance will be also the focus of future research. The research in this paper lays the foundation for future research.

## Figures and Tables

**Figure 1 sensors-23-02883-f001:**
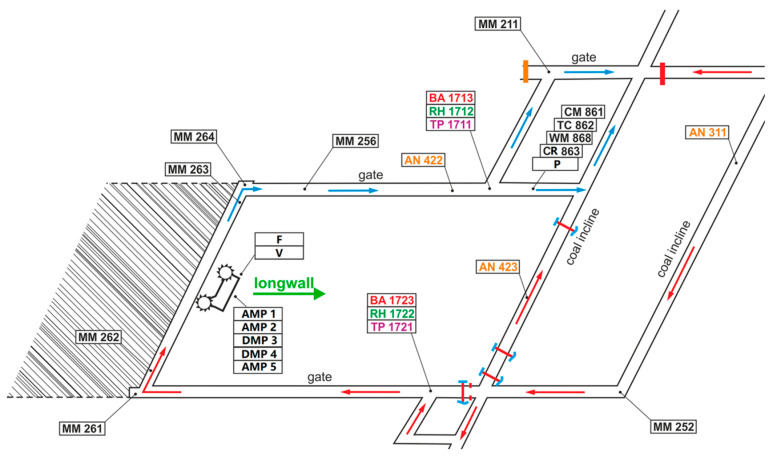
The layout of various sensors. The red arrows show the directions of air flow, and the blue arrows show the directions of gas flow.

**Figure 2 sensors-23-02883-f002:**
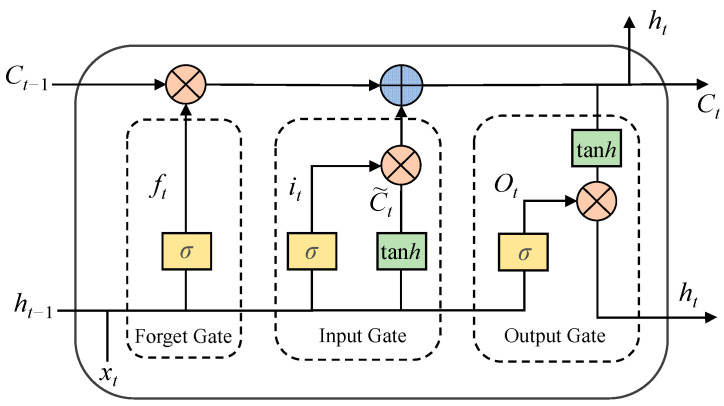
The basic structure of LSTM.

**Figure 3 sensors-23-02883-f003:**
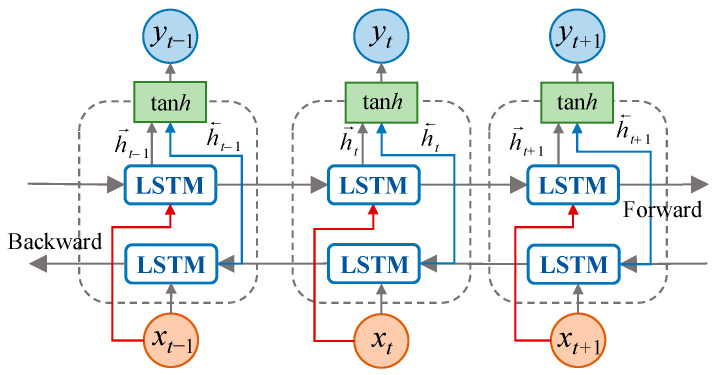
The structure of Bi-LSTM.

**Figure 4 sensors-23-02883-f004:**
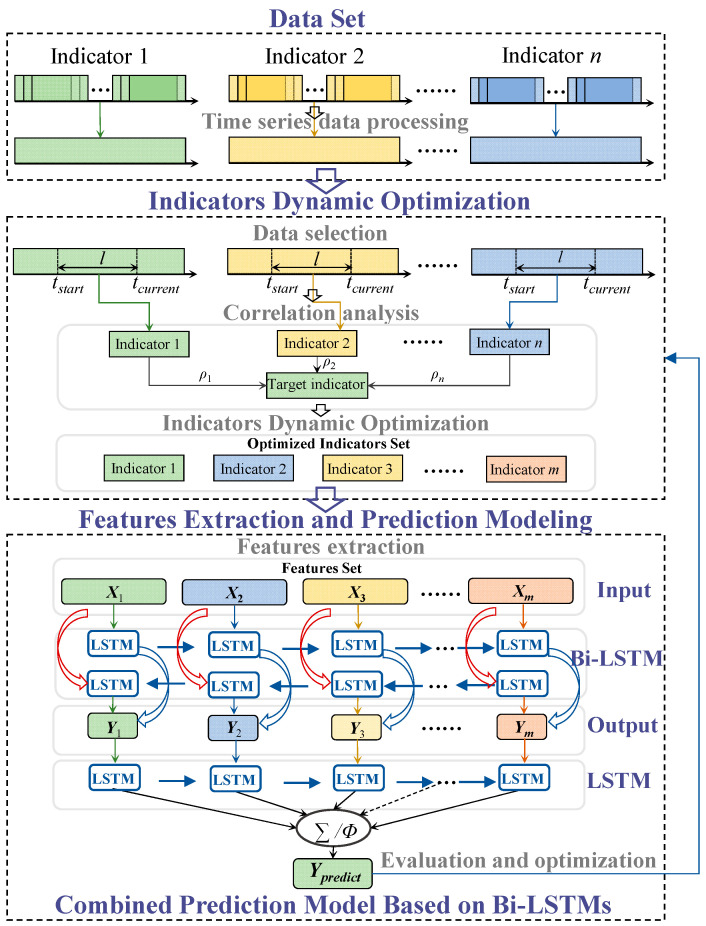
Combined prediction method of gas concentration based on indicators dynamic optimization and Bi-LSTMs.

**Figure 5 sensors-23-02883-f005:**
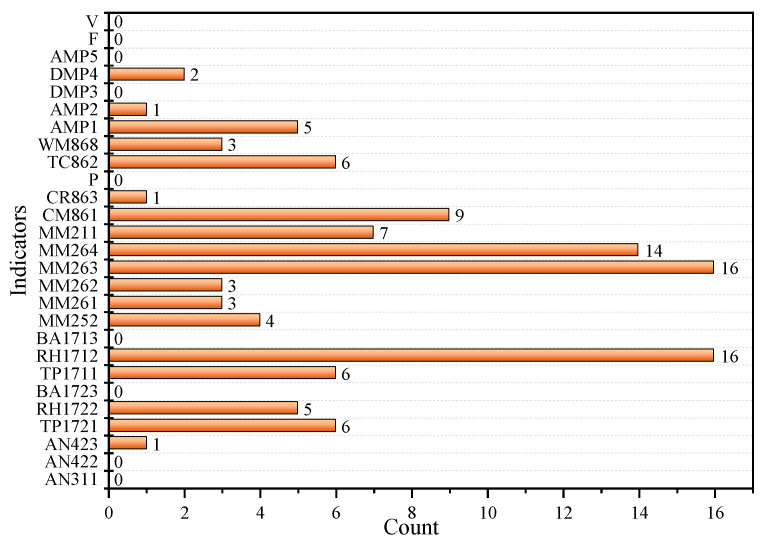
The frequency of each indicator in the top three indicators.

**Figure 6 sensors-23-02883-f006:**
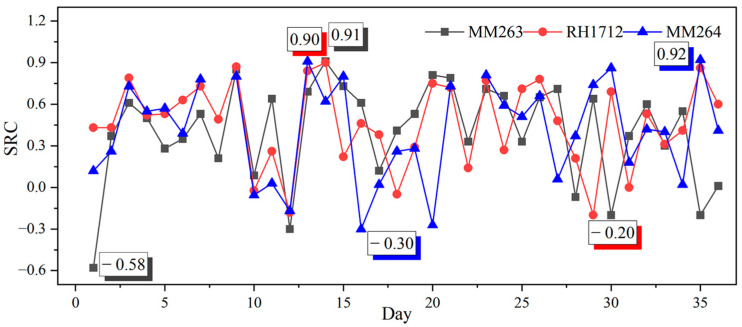
The change of SRC with time.

**Figure 7 sensors-23-02883-f007:**
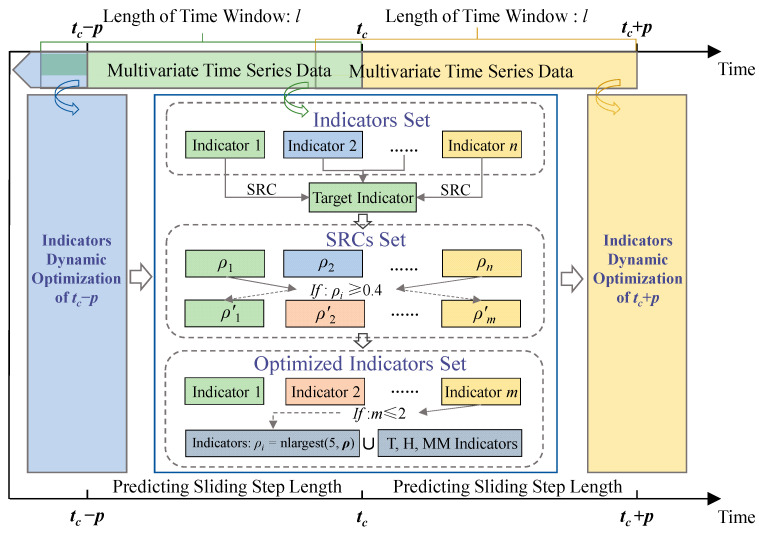
A dynamic optimization method for prediction indicators based on SRC.

**Figure 8 sensors-23-02883-f008:**
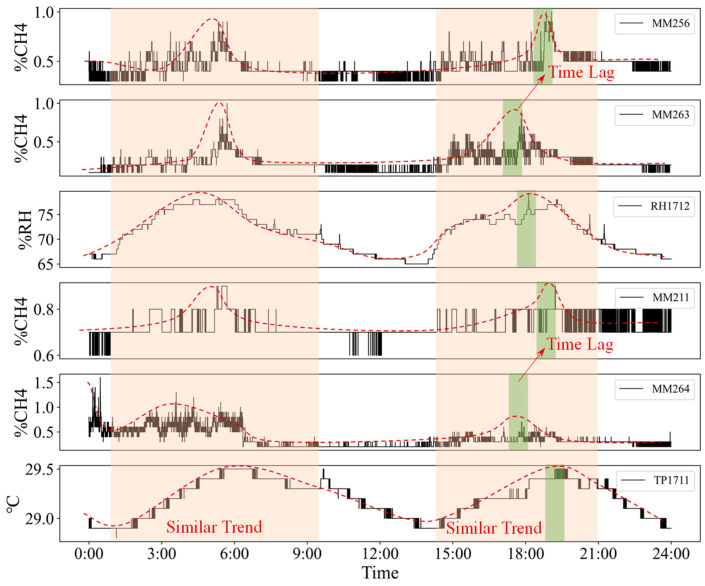
Time series monitoring data of different sensors.

**Figure 9 sensors-23-02883-f009:**
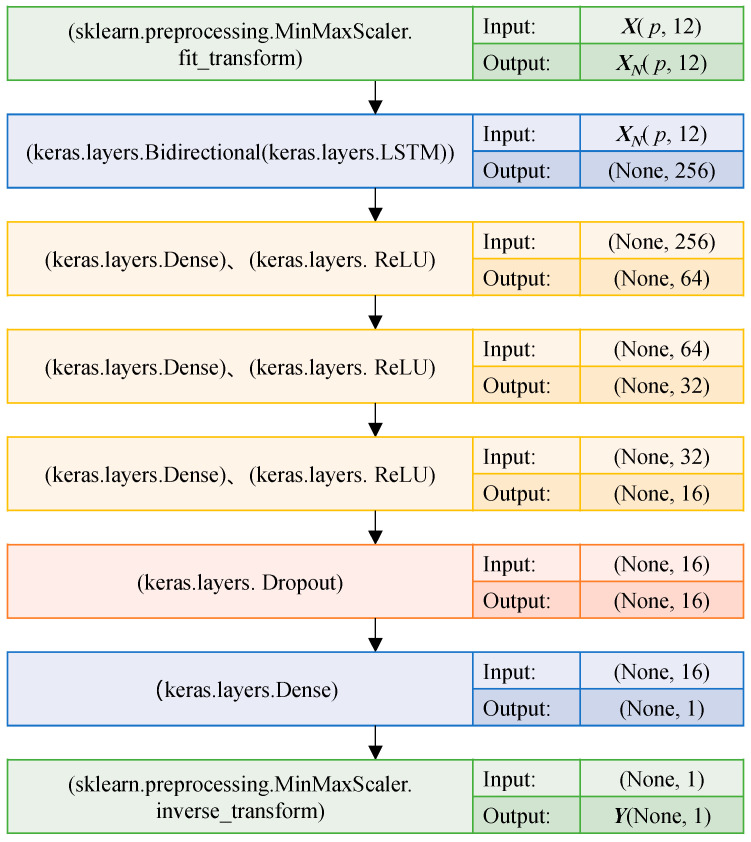
Bi-LSTM single indicator model network structure.

**Figure 10 sensors-23-02883-f010:**
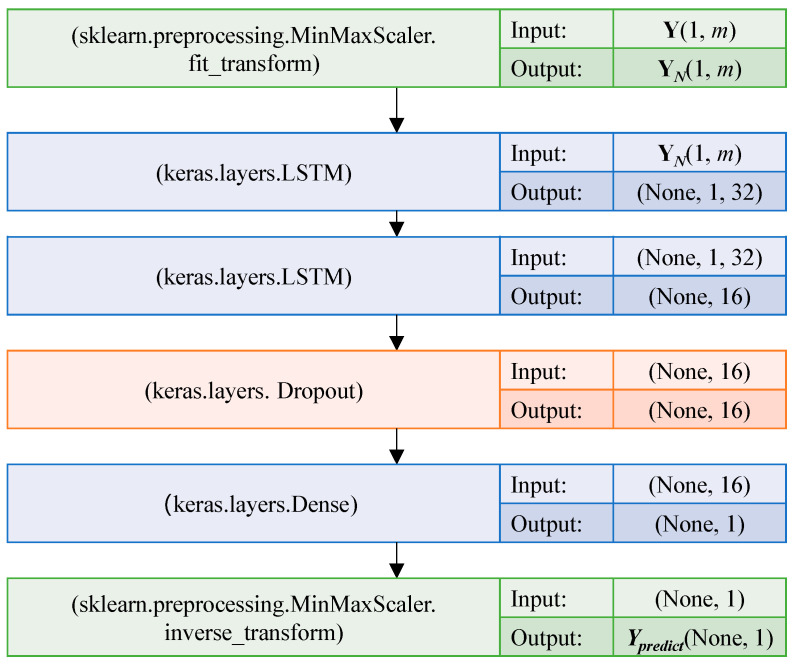
LSTM-optimized indicators combined model network structure.

**Figure 11 sensors-23-02883-f011:**
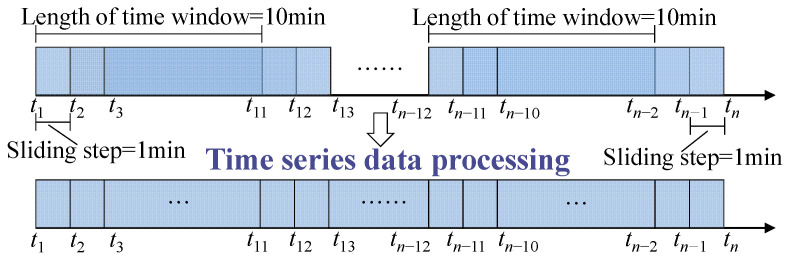
Time series data processing.

**Figure 12 sensors-23-02883-f012:**
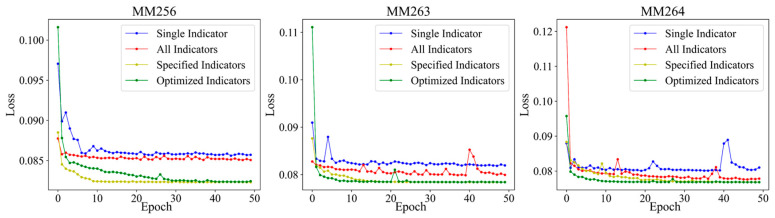
The changes of Loss with Epoch.

**Figure 13 sensors-23-02883-f013:**
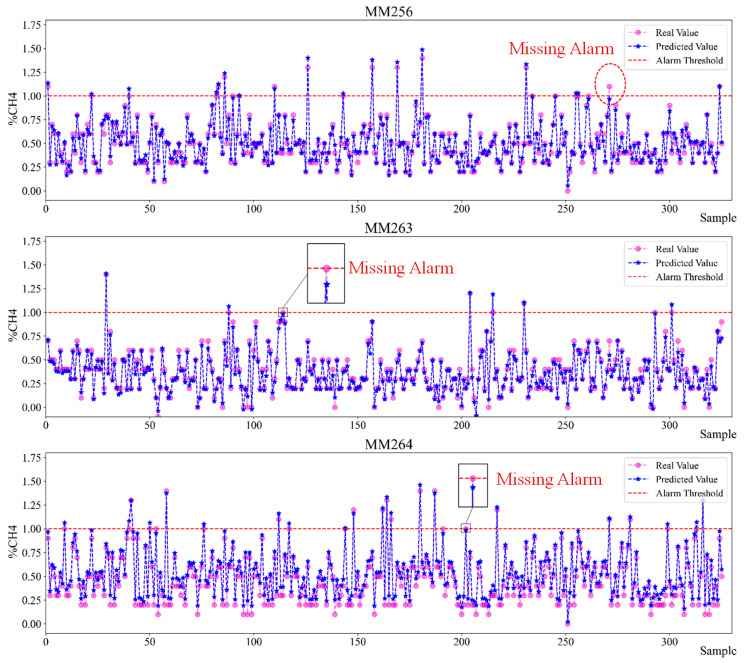
Prediction results of Bi-LSTMs dynamic optimized combined indicators model.

**Figure 14 sensors-23-02883-f014:**
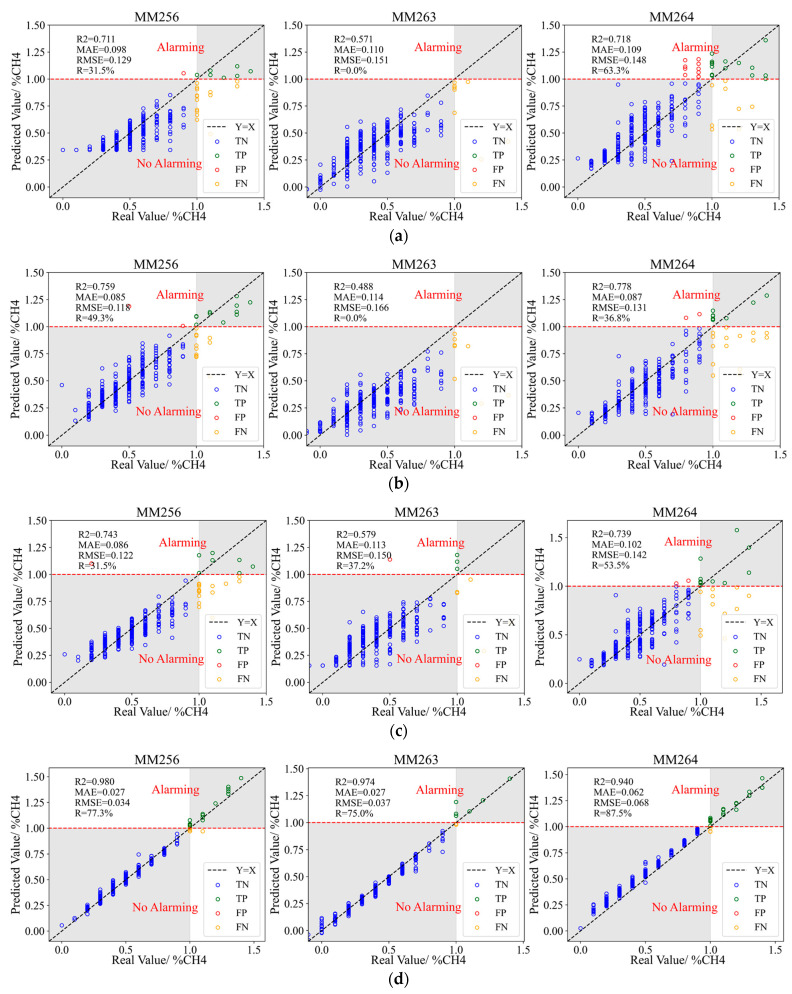
Prediction results of different indicators models. (**a**) Bi-LSTM single indicator model. (**b**) Bi-LSTMs all-indicators combined model. (**c**) Bi-LSTMs specified indicators combined model. (**d**) Bi-LSTMs dynamic optimized indicators combined model.

**Table 1 sensors-23-02883-t001:** The top three indicators of the strongest correlation with MM256.

Day	The Top 1 Indicator and SRC		The Top 2 Indicator and SRC		The Top 3 Indicator and SRC	
day 22	MM263	0.79	TP1711	0.76	TC862	0.73
day 23	MM262	0.51	MM252	0.45	MM211	0.34
day 24	MM264	0.81	RH1712	0.77	MM263	0.71
day 25	MM263	0.66	MM264	0.59	MM211	0.58
day 26	RH1712	0.71	MM264	0.51	AMP1	0.5
day 27	RH1712	0.78	MM264	0.66	MM263	0.65
day 28	MM263	0.71	AMP1	0.49	RH1712	0.48

**Table 2 sensors-23-02883-t002:** The evaluation of gas concentration prediction results by different indicators models.

Indicator(s)	Sensor	*R^2^*	*MAE*	*RMSE*	*R*^2^′	*MAE*′	*RMSE*′
Single Indicator	MM256	0.711	0.098	0.129	0.667	0.156	0.143
	MM263	0.571	0.110	0.151			
	MM264	0.718	0.109	0.148			
All Indicators	MM256	0.759	0.085	0.118	0.675	0.095	0.138
	MM263	0.488	0.114	0.166			
	MM264	0.778	0.087	0.131			
Specified Indicators	MM256	0.743	0.086	0.122	0.687	0.100	0.138
	MM263	0.579	0.113	0.150			
	MM264	0.739	0.102	0.142			
Optimized Indicators	MM256	0.980	0.027	0.034	0.965	0.039	0.046
	MM263	0.974	0.027	0.037			
	MM264	0.940	0.062	0.068			

**Table 3 sensors-23-02883-t003:** The evaluation of gas abnormal or normal emission prediction results by different indicators models.

Indicator(s)	Sensor	*TP*	*FN*	*TN*	*FP*	*FAR*	*MAR*	*R*	*FAR*′	*MAR*′	*R*′
Single Indicator	MM256	7	15	302	1	0.3%	68.2%	31.5%	1.2%	67.2%	31.6%
	MM263	0	8	317	0	0.0%	100.0%	0.0%			
	MM264	16	8	291	10	3.3%	33.3%	63.3%			
All Indicators	MM256	11	11	301	2	0.7%	50.0%	49.0%	0.4%	70.8%	28.7%
	MM263	0	8	317	0	0.0%	100.0%	0.0%			
	MM264	9	15	299	2	0.7%	62.5%	36.8%			
Specified Indicators	MM256	7	15	302	1	0.3%	68.2%	31.5%	0.4%	58.8%	40.7%
	MM263	3	5	316	1	0.3%	62.5%	37.2%			
	MM264	13	11	299	2	0.7%	45.8%	53.5%			
Optimized Indicators	MM256	17	5	303	0	0.0%	22.7%	77.3%	0.0%	20.1%	79.9%
	MM263	6	2	317	0	0.0%	25.0%	75.0%			
	MM264	21	3	301	0	0.0%	12.5%	87.5%			

## Data Availability

The processed data cannot be shared at this time as the data also forms part of an ongoing study.

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
