# Peer review of "Combined Prediction Model of Gas Concentration Based on Indicators Dynamic Optimization and Bi-LSTMs"

_sensors, 2023, doi:10.3390/s23062883_

Round 1

Reviewer 1 Report

This manuscript presents a model of for prediction of the gas concentration. This work is quite interesting to broad readership and will be useful. In my opinion, it can be accepted for publication after revision/clarification as suggested below;

- The type of sensors are very important. Please add the details of gas sensors such as type of gas sensors, target gas, concentration range…

- The temperature and humidity always affect to gas sensors. Please add the details of temperature and humidity sensors and correction models of humidity.

- All abbreviation of indicators should be identified in the full details such MM263, MM264, MM256, TPXXX,….

- The photograph of real detection system should be included in the manuscript.

Reviewer 2 Report

The paper presents a combined gas concentration prediction model based on dynamic optimization of indicators and Bi-LSTMs.

The authors highlight well the purpose of this work, detail well the stages of the presented study. The work is well documented, and the presented results support the performance of the presented model.

Reviewer 3 Report

Dear Authors, 

After reviewing your paper, I found it interesting. However, major revisions are recommended. More precisely English language, paper structure, and formatting should be addressed. Please see the attached document for details.

Kind regards,

Reviewer

Round 2

Reviewer 3 Report

Dear Authors,

It is nice to see such an improvement of a manuscript. The revised version of the paper addressed the noted suggestions. Overall, good job. 

I have no further comments.

Kind regards,

Reviewer